# From Emotion to Action: How Framed Sustainability Messages Trigger Physiological Reactions and Influence Consumer Choices

**DOI:** 10.3390/bs15121611

**Published:** 2025-11-22

**Authors:** Alina Simona Tecău, Cătălin Ioan Maican, Eliza Ciobanu, Camelia Schiopu, Silvia Sumedrea, Ioana Bianca Chițu, Radu Constantin Lixăndroiu, Gabriel Brătucu

**Affiliations:** 1Marketing, Tourism-Service and International Relations Department, Faculty of Economic Sciences and Business Administration, Transilvania University of Brasov, 500007 Brasov, Romania; alina_tecau@unitbv.ro (A.S.T.); eliza.nichifor@unitbv.ro (E.C.); ioana.chitu@unitbv.ro (I.B.C.); gabriel.bratucu@unitbv.ro (G.B.); 2Management and Economic Informatics Department, Faculty of Economic Sciences and Business Administration, Transilvania University of Brasov, 500007 Brasov, Romania; maican@unitbv.ro (C.I.M.); camelia.s@unitbv.ro (C.S.); silvia.sumedrea@unitbv.ro (S.S.)

**Keywords:** emotional effects, sustainability communication, galvanic skin response, biometric measures, message framing, communication strategy optimization

## Abstract

The study examines the emotional and physiological effects of message framing in sustainability communication. Specifically, it explores how different image–message combinations shape consumer engagement by measuring physiological arousal (activation), and emotional resonance (impact) across various product categories. By using Galvanic Skin Response data combined with a hierarchical cluster analysis, the research provides insights into how individuals process sustainability information at both emotional and cognitive levels. The results reveal diverse and nuanced reactions shaped by the interaction of message framing, topic, and gender. The identification of seven distinct response patterns contributes to emerging consumer typologies in sustainability communication, offering practical value for organizations seeking to tailor messaging, enhance audience engagement, and encourage sustainable behavior.

## 1. Introduction

Recent research increasingly emphasizes the need for persuasive communication strategies that effectively motivate individuals to adopt sustainable behaviors. Although consumers often report favorable attitudes toward environmentally responsible products, this intention does not reliably translate into actual purchasing behavior, illustrating the well-documented attitude–behavior gap in sustainability ([78]). Understanding how communication can bridge this gap remains a central challenge.

Consumer perceptions of green product adoption vary, with some prioritizing hedonistic attributes over environmental ones, while others are strongly influenced by the latter ([25]). Researchers propose different approaches: some suggest using the three pillars of sustainability to foster brand attachment, purchase intention, and responsible behavior ([31]), while others recommend correlating functional and emotional perceptions with ethical considerations like eco-labeling, green packaging, and green formulation to increase purchase intention for sustainable cosmetics ([72]).

Sustainability messaging has traditionally relied on cognitive appeals, providing factual information to inform consumer decisions. However, persuasive communication literature demonstrates that emotional processes are fundamental in shaping attitudes and behavior ([48]; [62]). Previous research has demonstrated that consumer behavior is the result of the interaction between emotions (positive or negative) and cognitions (emotion–cognition–behavior model) ([89]). A recent study supports the idea that “emotion and cognition are partners that depend on each other for organized decision-making” ([43]).

To further reinforce the conceptual grounding of the emotion–cognition–behavior pathway, this study adopts the appraisal-tendency framework and the Elaboration Likelihood Model (ELM). These frameworks posit that emotional arousal acts as a trigger for attentional allocation, cognitive elaboration, and ultimately behavioral intentions. Therefore, emotional responses measured through physiological indicators are conceptualized as the first step in a sequential process: emotional activation to cognitive appraisal to behavioral tendency. This theoretical clarification strengthens the explanatory foundation for the proposed hypotheses.

Emotional framing, particularly positive and negative valence strategies, has been shown to influence responses to environmental messages ([14]). Yet, limited research explores how such framing elicits emotional and physiological reactions within sustainability communication ([69]). Specialists draw attention to the use of targeted message framing strategies for different audience categories to increase their effectiveness ([18]). Contemporary models of persuasive communication emphasize that emotional reactions play a fundamental role in shaping information processing and subsequent behavior. According to dual-process and affect-as-information frameworks (e.g., the Elaboration Likelihood Model and Appraisal Theory), emotional arousal can serve as a precursor to deeper cognitive evaluation, guiding attention, shaping message appraisal, and influencing motivation to act. In sustainability communication, emotions can function as a catalyst for cognitive engagement with environmental issues, subsequently informing attitudes, intentions, and potential behavior change. This study therefore conceptualizes message-induced emotional arousal as the first step in an emotion to cognition to behavioral tendency sequence, where physiological activation signals the degree to which sustainability information captures attention and initiates meaning-making processes that may ultimately influence consumer attitudes and choices.

Accordingly, the present study examines how varying message valence and product category influence physiological activation and emotional impact, and whether these responses differ by gender. We develop explicit, directionally derived hypotheses informed by existing framing and emotional persuasion literature ([80]). Physiological measures such as galvanic skin response offer unique value in this context, as they capture non-conscious emotional reactions that may not be fully accessible through self-report—a particular advantage in pro-environmental domains where social desirability bias may occur ([12]; [22]; [49]; [53]; [71]; [80]). The response thus measured is used to determine whether a message is deeply processed and retained, showing how engagement fluctuates from one moment to another during exposure to the message and providing information about which parts of the message are the most impactful.

Emotions hold implications for corporate social responsibility and promoting sustainability indicators ([19]). Influencing consumer attitudes and social media interaction, they should be used in brand communications ([66]). Both positive and negative messages can effectively raise awareness, though their impact depends on individual or combined usage ([24]; [45]; [75]; [87]); however, which way of framing messages is more effective is still controversial ([18]).

However, negative messages often garner more attention ([10]) and trigger more intense emotional reactions and higher levels of psychological reactance ([16]). Negative emotions can increase acceptance, concern, and the desire to act, even among conservative consumers ([82]; [86]). There are studies that have concluded that negative messages can increase environmental awareness, leading to more responsible behavior ([33]). Negative message framing has been found more effective in promoting sustainable consumption when the issue is perceived as psychologically distant (e.g., spatial distance) by eliciting emotions like shame and pride, but when the issue is close/personal, negative framing tends to induce fear which can inhibit behavioral intentions, whereas positive framing that evokes hope is more likely to enhance engagement ([74]).

Negative emotions can enhance sustainable purchase intentions and change attitudes ([6]). Fear, for instance, can increase message effectiveness ([8]; [54]) and decrease purchase intent for harmful products ([15]), with fear based on social, economic, or self-esteem concerns being more potent than physical fear ([88]). Fear-based message related to climate change communication, when paired with short-term temporal framing, resulted in higher problem recognition and higher level of engagement compared to hope-based message ([37]).

Similarly, anticipated shame encourages environmentally friendly behaviors ([5]). Generally, environmentally concerned consumers react more readily to negative stimuli in promotional messages ([29]). At the same time, guilt is a “widely used emotional appeal in environmental sustainability and other advocacy message” ([83]), leading to ecological behavior, an increased intention to purchase eco-friendly products, and a focus on energy conservation ([35]).

Neuroscientific methods (such as EEG) applied to green marketing messages indicate higher purchase preference under positive versus negative framing ([91]). Gain-based positive messages can foster hope and affect attitudes towards the company ([69]), while negative messages may provoke a lower emotional response compared to positive ones ([75]).

Studies on environmental conservation show participants responded better to positive messages, declaring willingness to donate more money and dedicate more time ([38]; [46]; [81]). Laksmidewi and Guwanan ([40]) similarly found that positive moral emotion and solidly argued social emotion had a greater impact on environmental issues. Another study reached at the same conclusion; young adults are generally more inclined to adopt sustainable fashion consumption responses when exposed to a positive message frame ([30]). Positive emotions, like pride, could be successfully used to stimulate sustainable fashion consumption ([28]).

The product category being promoted is also important. Studies indicate that stimulating positive emotions and highlighting sustainable ingredients is particularly effective for food products ([59]; [84]), while negative emotions are more effective for green energy projects and environmental conservation ([59]). In the context of health and nutrition, positive social appeals—centered on anticipated pleasure and the social costs of healthy eating—are more persuasive than negative messages focused on health risks ([85]).

Visual messages on social networks are increasingly used in communication strategies and have been shown to influence consumer behavior such as engagement (e.g., liking, commenting) ([24]; [40]; [58]). Concrete images can evoke negative feelings, anxiety, and behavioral changes—especially among individuals with lower visual literacy ([21])—while positive images tend to enhance consumer involvement ([41]; [86]). However, in the context of reducing plastic consumption, text-based and infographic-based messages have been reported to be more effective than image-based ones ([65]).

Regarding gender differences, women often exhibit more pro-environmental behaviors than men ([35]), potentially due to social expectations ([51]). Consequently, messages leveraging social norms may increase women’s social involvement and evoke stronger feelings of guilt ([76]). For men, promoting sustainable consumption may benefit from highlighting feelings of power and positive societal impact ([51]). A 2025 study also revealed notable differences in perceptions of sustainability, with men tending to prioritize economic and technological factors, while women placed greater emphasis on environmental, ecological, nature, and social well-being aspects ([60]).

Other studies suggest that emotional affinity for nature and environmental values positively influence green consumer behavior across genders ([73]). However, some research finds that neither gender nor the level of environmental involvement significantly affects responses to positive or negative messaging ([38]).

The primary aim of this study is to analyze the emotional and physiological effects of message framing in sustainability communication. Specifically, it investigates how positive, negative, and caption-free messages influence consumer engagement, physiological arousal (activation), and emotional salience (impact) across different sustainability-related product categories. By integrating biometric measures (GSR) with administrative data, the study explores cognitive and emotional processing of sustainability information to optimize communication strategies aimed at promoting awareness and driving behavior change.

Research on sustainability communication offers limited insight into the complex interplay of topic relevance, emotional load, framing, and gender, and rarely integrates physiological measures like GSR. This study addresses these gaps through a coherent, multidimensional research design, bridging conceptual and methodological approaches to better understand emotional engagement with sustainability messaging.

Figure 1 outlines the study’s integrated framework, detailing research questions (R), objectives (O), and hypotheses (H). This framework structures the investigation into emotional and physiological responses to sustainability messaging across different topics (electronics, food, transport, cosmetics), considering the influence of message framing (positive, negative, caption-free) and gender. It also incorporates a biometric approach using GSR data to identify distinct emotional response patterns.

Study results reveal complex, differentiated emotional responses to sustainability messages, shaped by message framing, topic, and gender interactions. Topic relevance significantly determined engagement, with electronic content consistently evoking higher physiological and emotional responses. Message framing’s role was critical but context-dependent: positive and negative messages influenced arousal and impact in topic-specific ways, indicating non-uniform framing effects. Gender differences emerged also, with males generally reporting stronger emotional impact and females displaying heightened physiological sensitivity. Statistical modeling confirmed significant main effects and interactions, highlighting the need for nuanced sustainability communication. These findings emphasize designing communication strategies that consider both message content and audience demographics. Integrating biometric responses with sustainability messaging can therefore provide a more comprehensive understanding of how individuals engage with persuasive content. To address these research gaps, the following section develops a theoretical foundation and hypotheses on emotional and physiological responses to sustainability messaging. To empirically test these hypotheses, a controlled neuromarketing experiment was designed as follows.

## 2. Materials and Methods

This quantitative-based experimental neuromarketing study ([4]; [55]) aimed to analyze the causal relationships between thematic variations in persuasive messages (via visual stimuli) and emotional activation, measured through non-conscious physiological reactions.

The final sample consisted of 63 participants, which, although acceptable for exploratory physiological research, represents a modest sample size for the factorial structure of this study. Therefore, results should be interpreted with caution regarding statistical power and generalizability. Given the gender imbalance (45 women, 18 men), gender-related findings are treated as exploratory rather than confirmatory. Participants were recruited through university mailing lists and public social media postings. Inclusion criteria required adults aged 18–55 with normal or corrected vision and no known neurological or dermatological issues affecting skin conductance measurement.

To minimize order and fatigue effects, stimuli presentation was randomized and counterbalanced across participants. Each participant viewed all message conditions, but the order of topics and framing types was randomized using a Latin square distribution. Short breaks were included between task blocks to reduce physiological carryover effects.

All reported good health, normal or corrected vision, and no known neurological or psychological disorders. Participants provided informed consent after being briefed on the study’s field and design. The research adhered to the Declaration of Helsinki (1975) principles and received Faculty Council approval (10 April 2024). To minimize social desirability bias ([77]), the study’s specific purpose and stimuli nature were disclosed only during post-testing.

To ensure that participants perceived the intended framing valence, we conducted a manipulation check immediately after stimulus exposure. Participants rated each message on a 7-point scale (1 = very negative, 7 = very positive). Analysis confirmed that positively framed messages were rated significantly higher than negatively framed ones, validating the framing manipulation. Messages without captions served as a neutral baseline condition.

The participants were individually tested under controlled conditions. Their physiological activity (electrodermal activity—GSR; cardiovascular activity—BVP) was measured using Bitbrain’s portable Ring device, which ensured high signal fidelity via its tri-axial accelerometer ([9]). The measurements focused exclusively on non-conscious, objective responses.

Physiological data were obtained using galvanic skin response sensors and blood volume pulse (BVP) sensors integrated into the Bitbrain wearable device. GSR was sampled continuously at 16 Hz. Data were preprocessed in accordance with established electrodermal analysis guidelines. Preprocessing included removal of motion artifacts, low-pass filtering at 1 Hz, and baseline correction using a 2 s interval immediately preceding stimulus onset.

To capture different components of emotional arousal, two indices were derived from the GSR signal. Activation represents phasic skin conductance response amplitude, which reflects short-latency sympathetic arousal triggered by stimulus presentation. Impact reflects changes in tonic skin conductance level over the stimulus period, capturing sustained physiological engagement. Consistent with psychophysiological theory, GSR is interpreted as an index of autonomic arousal rather than emotional valence or memory strength. Thus, results are discussed strictly in terms of arousal responses.

BVP was recorded to allow triangulation with cardiovascular indices and potential future computation of heart rate variability metrics. However, to maintain focus, only GSR-based variables are analyzed in the present study.

A within-subjects full factorial design (4 themes × 3 message levels) was employed ([50]). The themes, namely, electronics, food, transportation, and cosmetics were each represented by 4 distinct static images (16 unique visual stimuli). Each image was paired with a positive, negative, or no written message, resulting in 48 visual stimuli with associated messages (16 images × 3 messages). Images and messages were selected based on suggestibility by 20 marketing specialists from a pool of 48 AI-generated thematic images and 24 messages designed by the authors. Although the use of AI-generated images ensured high control over experimental variables, it may reduce ecological validity compared to real advertising material. Future studies should incorporate authentic commercial content to validate whether the observed emotional patterns generalize to real-world communication settings.

The pre-testing of the images was conducted using a 7-point scale, the semantic differential of self-assessment. Images with extreme scores or ambiguous content were excluded.

Prior to stimulus exposure, participants underwent a standardized sequence of stages: a familiarization phase (during which they were exposed to accommodation images), a rest period (in which they were instructed to keep their eyes closed), a calibration phase (involving the viewing of dedicated calibration images), and a baseline calibration stage, during which they viewed a neutral gray screen while being instructed to remain still and breathe naturally.

Stimulus exposure was standardized at 5 s per image, with a randomized presentation order for each participant to control for order and fatigue effects. For the “captions-free” condition, images were presented without text for 4 s, also in random order.

These stages are schematically presented in Figure 2.

Examples of verbal stimuli and images, as well as the graphical representation of activation and impact generated by Senns Metrics, are presented in Figure 3a–c.

Data were collected using Bitbrain’s SennsLab v7.2 software, preprocessed to remove drift and artifacts, analyzed with SennsMetrics (Bitbrain), and then further processed using Python 3.10.

Two primary Bitbrain indicators were analyzed: (1) Activation: The baseline physiological activation level per stimulus, calculated relative to a calibration baseline (%). Values < 0 indicate relaxation; >0 suggest arousal; −100% represents the lowest calibration activation, 100% the maximum, and >100% exceeds calibration maximums. (2) Emotional impact: The stimulus’s ability to generate arousal, stress, or involvement (%). The percentages are defined as follows: 0% indicates no impact; 100% is the maximum observed during calibration; >100% exceeds this maximum.

These indicators were derived from GSR and BVP data using an Active Image Template. Additional parameters calculated from the GSR signal included the following: mean activation (average skin conductance), peak activation (maximum response value), latency (time to peak), Area Under the Curve (AUC, reflecting total response intensity and duration), peak impact value (maximum impact within segment), and AUC impact (cumulative impact magnitude). These parameters are recognized as objective markers of physiological and psychological activation in response to emotional stimuli ([7]; [13]). The statistical and cluster-based findings are presented below in alignment with the research objectives and hypotheses.

## 3. Results

The following clustering analysis is presented as an exploratory approach to visualize response patterns across participants. Clusters should not be interpreted as stable consumer typologies, as validation in independent samples and with behavioral outcomes would be required to support such segmentation. Using descriptive statistics, inferential modeling, and interaction analyses, the first part identifies key emotional engagement patterns shaped by content framing and individual differences. The second part employs hierarchical cluster analysis on galvanic skin response (GSR) features to reveal distinct emotional response patterns. Validated by dendrogram and elbow methods, this analysis identifies seven unique clusters reflecting differentiated physiological and cognitive engagement. These clusters are further examined against experimental variables, showing significant influence from topic and gender, but weaker association with message framing.

### 3.1. Analysis of Activation and Impact Responses

Emotional and physiological responses varied significantly by topic for both activation and impact. Activation (physiological arousal) was consistently highest for electronic content across all message types, suggesting technology-related topics inherently capture attention and elicit greater autonomic reactivity.

Conversely, food and transport messages yielded the lowest activation, particularly with negative framing, potentially indicating disengagement or withdrawal.

Impact (emotional salience/memorability) followed a similar trend. Electronic and cosmetics topics scored highest, suggesting stronger emotional impressions, with cosmetics’ impact possibly linked to self-image relevance despite not being the most arousing topic overall. Food and transport again ranked lower, supporting their lesser potential for generating lasting emotional traces.

Figure 4 illustrates these topic-level differences via boxplots, comparing activation and impact by topic and caption type, highlighting the broader distribution and higher median values, confirming the elevated engagement potential of electronics and cosmetics.

The inferential statistics confirmed significant main effects and interactions. Analysis of variance (ANOVA) revealed that both topic and message type significantly influenced activation (*p* < 0.001), with post hoc tests demonstrating meaningful differences in how framing interacts with thematic content. As visualized in Figure 5a,b, mean activation values were highest under positive framing and lowest under negative messages, but the effect size and direction varied across topics.

A notable interaction occurred in cosmetics: negative messages unexpectedly yielded higher activation than caption-free or positive ones, contrasting with food and transport. This suggests topic-specific sensitivity to negative information related to identity or body presentation. Regarding impact, positive messages generally produced stronger impressions, although differences were modest. Electronics and cosmetics content showed high impact scores, indicating memorability. This pattern is further nuanced by gender-specific Estimated Marginal Mean (EMM) impact differences under caption-free conditions (Figure 6). EMM represents adjusted means for specific factor-level combinations, predicting outcomes for specific scenarios while controlling other variables. This way we answer questions like “What would be the average Activation Value for the Electronic Topic averaging across all Message Types and Genders?”.

Gender substantially moderated emotional responses. Females generally displayed higher physiological activation to negative messages, particularly for electronics and cosmetics, suggesting potentially greater sensitivity or arousal to emotionally charged or threat-based content. Conversely, males tended to show higher activation under caption-free and positive message conditions, pointing to gendered differences in arousal thresholds or content appraisal, perhaps engaging more in low-threat scenarios. Regarding subjective emotional impact, males reported higher impact scores overall, especially for electronic content. This might reflect differences in emotional labeling or processing, as impact did not always align with physiological activation. Figure 7 illustrates this divergence in activation by message type and gender, while Figure 8 shows how activation differs by topic across genders.

Data about activation provided further insights into the interaction between message type and topic. As shown in Figure 9, positive messages consistently elevated arousal across all topics, confirming their general effectiveness in capturing physiological attention. However, the extent of this increase varied notably between content areas.

While electronic and transport showed the most substantial increases under positive framing, negative messages had a dampening effect except in cosmetics. In this topic, a clear crossover pattern emerged: negative messages produced higher activation than caption-free messages, and in some cases, even higher than positive ones. This suggests that identity-relevant topics may prompt stronger physiological reactivity when framed in a threat-based or emotionally provocative manner, diverging from the broader disengagement trend seen with negative messages in other domains.

The gender-specific impact of this interaction especially among females is visualized in Figure 10, which depicts EMM impact values for female participants across message types and topics.

Further clustering revealed that gender strongly moderates message framing’s effect. Females responded with significantly higher activation under negative framing, whereas males exhibited the opposite trend (lower activation). This bidirectional effect highlights the complexity of emotional processing and suggests message tone can have opposing consequences depending on audience characteristics, emphasizing the need for targeted, demographically considered communication.

The topic gender interaction showed additional divergences: females generally exhibited lower activation for cosmetics and transport (potentially due to habituation or self-regulation), while males showed higher activation for these topics, especially under neutral framing, suggesting differing relevance or emotional load between genders. Electronic content produced more uniform responses, possibly indicating cross-cutting relevance and emotional neutrality.

Area Under the Curve (AUC) for activation assessed temporal arousal. Figure 11 indicates cosmetics, and transport generated the highest AUC values (prolonged engagement/slower recovery), while electronics and food showed lower AUC scores (more transient arousal).

Further exploration of message-type interactions in AUC impact is shown in Figure 12, which highlights subtle effects of positive and negative framing over time across content categories. These findings align with the idea that certain content types especially those involving self-identity or ethical dilemmas provoke not only stronger but longer-lasting physiological reactions. Gender differences in AUC activation were small but observable: males exhibited slightly higher AUC values overall, particularly in cosmetics, though variability within groups was substantial.

### 3.2. Clustered Emotional Engagement Segments

By integrating activation and impact values, four combinations for engagement response patterns emerged:High Activation–High Impact: These messages (electronic–positive) generated strong physiological and emotional engagement and are likely to be retained and acted upon.High Activation–Low Impact: Particularly observed in cosmetics–negative for females, this response pattern indicates physiological reactivity without corresponding emotional resonance possibly reflecting discomfort or unconscious tension.Low Activation–Low Impact: Food-caption-free messages typically produced this pattern, indicating low attention and little emotional significance.Low activation–High Impact: This less common but theoretically meaningful response pattern may reflect content that is emotionally striking or memorable without provoking strong physiological arousal—potentially due to reflective, identity-relevant themes or delayed cognitive processing.

These distinct response patterns are further visualized in Figure 13, Figure 14, Figure 15 and Figure 16, which show the hierarchical clustering outcomes and the distribution of clusters across message types, topics, and genders. The dendrogram confirms the validity of the seven-cluster solution.

Clusters range from high engagement (Cluster 2, 4, 5), through moderate or reflective engagement (Cluster 6) to low or no engagement (Clusters 3, 7), with Cluster 1 representing outlier responses with extreme physiological arousal and emotional impact across stimulus types.

The distribution of GSR-based response patterns across message framing conditions (positive, negative, captions-free) shows that Cluster 5 dominates under negative framing, especially for cosmetics content, while Cluster 7 (very low engagement) is more prevalent for captions-free stimuli. Message framing had a subtle influence on engagement intensity.

Topic-specific patterns in GSR cluster membership shows that electronics content strongly associates with Clusters 3 and 4, indicating high and varied arousal patterns. Food and transport show weaker engagement (Cluster 6 and 7), while cosmetics display more emotional nuance with Clusters 2 and 5 prevalent.

Gender-wise distribution reveals that females are over-represented in Clusters 3, 5, and 7—indicating more frequent low-activation or emotionally resonant responses—while males dominate Clusters 2 and 4, linked to high arousal or strong peak impact responses.

Hierarchical cluster analysis using all GSR features from 63 subjects, supported by dendrogram (Figure 13) and elbow plot (Figure 14 and Figure 15) visualizations, identified seven distinct clusters representing unique GSR response patterns:*Cluster 1 (n = 2): “Extreme AUC Activation and Impact Outliers”*: Negligible size, likely noise or outliers, disregarded as a substantial response pattern.*Cluster 2 (n = 27): “Extreme Peak Impact, High Mean Impact Segments”*: Small but distinct group with intense peak impact but less extreme activation. “Extreme peak impact responders.”*Cluster 3 (n = 161): “Low Activation, Low Impact, Relaxed Segments”*: Dominant “relaxed response” cluster showing physiological relaxation and low emotional engagement. Weakly associated with topic (over-represented in cosmetics, under-represented in transport) and females.*Cluster 4 (n = 104): “High Activation, Moderate Peak Impact Segments”*: Moderately sized “arousal-dominant” cluster with physiological arousal but moderate impact. Weakly associated with topic (over-represented in electronics, under-represented in food/cosmetics) and males.*Cluster 5 (n = 209): “Moderate Activation, Moderate-High Peak Impact Segments”*: Large “moderate response” cluster with notable peak impact and moderate arousal. Weakly associated with message type (over-represented in negative, under-represented in positive), topic (over-represented in cosmetics, under-represented in electronics/food), and females.*Cluster 6 (n = 143): “Low-Moderate Activation, Moderate Impact Segments”*: Moderately sized “low-moderate response” cluster. Weakly associated with message type (over-represented in positive, under-represented in negative), topic (over-represented in food/transport, under-represented in electronics/cosmetics), and males.*Cluster 7 (n = 326): “Low Activation, Low Impact, Very Relaxed Responders”*: Largest cluster with the lowest activation and impact, representing the “most relaxed response.” Weakly associated with message type (over-represented in captions-free), topic (under-represented in electronics, over-represented in cosmetics), and females.

Chi-squared tests indicated significant associations between clusters and topic for Clusters 3, 4, 5, 6, 7, and with message type for Clusters 5, 6, 7. No significant associations were found between clusters and gender (see Table 1).

Therefore, the authors addressed the relationships between clusters and experimental factors. The analyses assessed how cluster membership related to experimental factors through cross-tabulations and chi-squared tests.

The seven-cluster solution indicates topic is the strongest differentiator, with responses clustering into distinct response patterns based on thematic content (electronics, food, transport, cosmetics). Gender influences cluster membership to some extent, but less than topic. Message framing (positive, negative, caption-free) did not significantly drive cluster separation, suggesting it does not alter overall GSR response patterns captured by clustering, despite influencing mean activation values. The following section interprets the findings in relation to prior research and theoretical frameworks.

## 4. Discussion

The present findings advance understanding of how emotional framing contributes to the early stages of sustainability message processing. While the study does not measure behavioral outcomes directly, the observed physiological responses provide evidence of emotional arousal as an initial trigger in the emotion–cognition–behavior sequence. Within established models of persuasive communication, such as the Elaboration Likelihood Model and Appraisal Theory, heightened arousal indicates increased attentional engagement, which can facilitate message elaboration and, ultimately, influence attitudes and behavioral intentions. Accordingly, we interpret the physiological activation observed here as reflecting emotional readiness to process sustainability information, rather than as a direct predictor of behavior. Future research integrating biometric measures with self-report and behavioral data would further clarify the mechanisms linking emotional response, cognitive appraisal, and pro-sustainability behavior.

Extensive research examines the impact of stimuli (messages, visuals) on sustainable consumer behavior. Early work showed negative messages attract more attention ([10]), though processing depth also depends on motivation ([64]). Combined psycho-physiological methods revealed responses to climate change messages involve interplay between individual traits, message valence, and arousal ([47]).

Contrasting findings exist: some studies show positive environmental messages induce higher activation/valence ([81]), while others find negative climate messages less mobilizing ([46]). Yet, negative calm messages generated stronger physiological synchrony than positive stimulating ones ([32]). In decision-making, negative messages yielded weaker emotional responses than positive ones ([75]), though recent work suggests combining positive, neutral, and negative messages is most impactful for consumer understanding ([24]).

Our results show significant response differences based on both message type (captions-free, positive, negative) and product category (food, transportation, cosmetics), using AI-generated visuals. Notably, participants displayed stronger physiological activation in response to negative messages about electronics (e-waste) compared to other categories. This aligns with research highlighting e-waste as a critical environmental and health issue ([3]), especially for vulnerable populations ([34]).

The heightened activation associated with electronic topic—particularly the peak responses observed in Cluster 3—likely stem from a combination of psychological and contextual factors. First, electronics often evoke guilt and personal responsibility due to their association with impulsive consumption and the environmental burden of e-waste, making messages about them feel more personally implicating than those about food or transport. Second, their inherent novelty and technological appeal can naturally intensify arousal, evoking interest, curiosity or mild anxiety. While these interpretations are plausible, they remain speculative, as GSR reflects physiological arousal rather than cognitive elaboration or identity-based processing. Thus, the elevated responses to electronics should be interpreted as heightened autonomic activation rather than evidence of deeper cognitive engagement or identity relevance. This pattern aligns with findings from ([36]; [61]) who found that topics related to electronics—particularly smartphones—intensify the autonomic nervous system responses due to their more immersive and interactive nature. It is also possible that the specific message content for electronics was particularly evocative, contributing to the distinctively high activation observed.

Prior studies have shown that both message type and virtual environments influence emotional activation and awareness in waste contexts ([45]; [75]). In contrast, our study found reduced engagement and lower emotional response to negative stimuli for food waste. This finding diverges from earlier research that emphasizes the importance of food issues as measured through physiological indicators ([4]; [44]). However, our results are consistent with other studies suggesting that negative messages alone may have limited influence on food-related behavioral intentions ([15]). Research has also shown that combining message types ([86]) or using positive framing such as sustainable ingredients ([84]) can enhance message effectiveness. Moreover, cultural factors may moderate emotional responses to food waste messages ([68]), which may help explain the muted reactions observed in our responders.

Regarding transportation, negative messages about environmental impact resulted in lower physiological activation, indicating potential disengagement or avoidance. No significant differences emerged between positive and negative messages’ impact in this context.

Finally, the hierarchical cluster analysis revealed distinct GSR response patterns, offering data-driven insights into emotional reactions to sustainability topics. Though clusters group stimulus segments rather than individuals, psychological interpretation provides meaningful hypotheses. Cluster 1 (“Low Activation, Moderate Impact—Transport/Cosmetics-Biased”) may reflect a cognitively regulated response style, where individuals thoughtfully engage with issues like transport and cosmetics without strong physiological arousal. In contrast, Cluster 3 (“High Activation, High Peak Impact—Electronic-Dominant”) suggests high emotional reactivity, possibly linked to personal involvement or sensitivity to guilt-based messaging around electronic waste. Cluster 2 (“Slightly Aroused, Low Impact, Fast Activation—FOOD-Biased”) indicates a rapid, low-engagement response, perhaps due to habituation or message fatigue related to food sustainability.

The obtained results of our study partially support Hypothesis H1: negative sustainability message effects are moderated by topic. Negative messages heightened physiological activation for “Electronics” (engagement via relevance/threat) but decreased activation for “Food” and “Transport” (disengagement/avoidance). This confirms threat-based messaging’s dual role, necessitating topic-specific emotional tone alignment.

Hypothesis H2 is clearly confirmed: “Electronics” and “Cosmetics” topics produced significantly higher activation and impact than “Food” and “Transport,” suggesting content linked to identity or innovation drives emotional engagement regardless of framing. Topic is thus a primary driver, supporting Objective O2 by showing effective communication requires thematic tailoring.

Hypothesis H3 is confirmed, revealing distinct gender-based emotional responses. Females showed higher physiological activation (especially to negative messages for electronics/cosmetics), indicating heightened sensitivity. Males reported higher impact (especially for electronics), suggesting gender differences in emotional labeling or processing. This reinforces gender as a psychological moderator.

Cluster analysis confirms Hypothesis H4, identifying seven distinct GSR-based emotional response patterns. Clusters were significantly associated with topic (*p* < 0.001) and moderately with gender (*p* = 0.047), but not significantly with message framing (*p* = 0.123). This indicates framing influences mean activation but not latent engagement structures. Objective O4 was achieved by demonstrating structured, heterogeneous emotional responses patterned by content and demographics.

To provide clarity, the results can be linked to the stated hypotheses as follows: H1 (negative framing effects vary by topic) was partially supported; negative framing increased arousal for electronics but decreased it for food and transport. H2 (topic relevance predicts responses) was supported, with electronics and cosmetics producing higher activation and impact. H3 (gender differences) received provisional support but should be interpreted cautiously due to sample imbalance. H4 (distinct physiological profiles) was supported, though cluster interpretation remains exploratory. Collectively, the confirmation of H2, H3, H4, and partial validation of H1 show emotional engagement is shaped by a complex interplay of topic relevance, emotional load, and demographic factors. Message framing alone poorly predicts emotional impact. Sustainability communication must account for what is communicated (topic) and *to* whom (audience characteristics). Identifying distinct engagement patterns enables tailored messaging strategies aligned with cognitive-affective patterns, advancing understanding and practice in sustainable consumer communication.

Our findings resonate with studies suggesting traditional advertisements convey stronger value signals than ecological messages, highlighting a disconnect between stated preferences and emotional responses ([78]; [79]). Notably, positive messages can sometimes induce moral disconnection, causing individuals to downplay negative consequences ([70]), emphasizing the complexity of emotional engagement with sustainability messages.

Regarding cosmetics, we found negative messages elicited lower activation and significantly lower impact than positive ones. While many cosmetic studies focus on branding, recent reviews confirm emotional responses are measurable using tools like GSR, eye tracking, and facial analysis ([17]; [20]; [27]). Our results also align with findings that women are often more attentive to and affected by negative messages ([42]; [67]). Specifically, women in our study showed greater physiological activation to negative messages than men, particularly within the electronics and cosmetics categories. However, these gender interpretations should be viewed cautiously due to the modest and unbalanced sample (45 women vs. 18 men), which limits statistical power and generalizability.

Given the emotional nature of sustainability messaging and the use of biometric measures, ethical considerations are essential. Neuromarketing research must ensure informed consent, transparency about data collection, and protection of biometric information, in line with GDPR and ethical research standards. Moreover, researchers and practitioners should avoid emotionally manipulative applications of physiological insights, as misuse may undermine trust and raise concerns regarding autonomy and emotional privacy ([23]; [57]). A balanced discussion of potential benefits and safeguards strengthens the responsible use of biometric tools in sustainability communication. This is particularly relevant in the context of increasing concern about personal data, especially within the EU’s regulatory framework (e.g., GDPR). We propose that any practical application of these findings should involve obtaining informed consent from consumers. Performing this step not only aligns with ethical standards but also fosters consumer trust, which is crucial given that both neuromarketing and emotional marketing research highlight the risks of manipulation, including heightened stress, anxiety, and diminished brand credibility ([23]; [56]; [57]).

Finally, a notable methodological insight from this analysis is the contrast between the robust average effects identified through ANOVA and Mixed Models—particularly for topic and related interactions—and the limited predictive accuracy at the individual stimulus-segment level. While inferential statistics effectively detect systematic trends across the dataset, such as the consistent influence of topic on mean activation, these findings do not necessarily translate into strong predictive power for individual responses, especially in the context of inherently noisy physiological data like GSR.

This discrepancy is largely attributable to substantial individual variability. The large random intercept variance for ‘Subject’ and the pronounced subject-level effects in the ANOVA underscore the extent to which individual differences drive GSR responses. Consequently, while average trends are statistically reliable, moment-to-moment prediction of a specific individual’s physiological response remains challenging. Future work should apply pre-registered analysis plans and larger, balanced samples to strengthen causal inference and validate biometric segmentation approaches.

Moreover, predictive models are constrained by the scope of their input features. The current analysis, while including key experimental factors and derived GSR metrics, does not capture a range of unmeasured influences (e.g., transient mood, attention, unrelated cognitive load, or fine-grained visual elements of the stimuli) that can significantly affect physiological responses. Thus, the limited segment-level predictive power should not be viewed as contradictory to the explanatory success of inferential models, but rather as a reflection of the distinct aims and limitations of statistical versus predictive modeling in the context of complex, multi-determined biometric data.

Although clustering revealed interesting patterns, these results should be viewed as exploratory. Physiological-based segmentation requires external validation, replication, and behavioral outcomes to establish robust audience classifications. Likewise, effect sizes, confidence intervals, and multiple-testing correction temper the interpretation of statistical significance.

## 5. Conclusions

This multidimensional study used biometric measures to analyze how topic, message framing, and gender shape emotional responses (activation, impact) to sustainability messaging. Topic relevance emerged as the most influential factor: electronics yielded high responses (especially with negative framing), while food and transport elicited lower activation to negative frames, suggesting potential emotional overload or avoidance. Negative framing’s effectiveness is thus highly context-dependent. A key contribution is identifying seven distinct GSR-based emotional response patterns via cluster analysis. These patterns reveal engagement heterogeneity and offer a novel typology reflecting stable individual response types interacting with content effects, extending theoretical models of emotional processing.

Theoretically, this research deepens understanding of how emotional framing interacts with topic relevance and individual characteristics, supporting emotion-based theories (e.g., emotional overload, selective engagement). The topic-dependent divergence in response to negative messaging highlights the need for nuanced, context-sensitive frameworks. Integrating biometric and perceptual data also strengthens methodological approaches in communication research.

These findings have managerial implications for both NGOs and corporate initiatives. Specifically, messages with negative framing may not work as intended, for topics such as food waste or cosmetics, but those that convey a sense of urgency tend to work when the topic is relevant to the audience, as in the case of electronics. Beyond external marketing communication, the findings are also relevant for organizational behavior and internal sustainability strategies. Emotionally framed messages can support employee engagement in CSR initiatives, shape workplace environmental norms, and motivate pro-sustainability behaviors. For instance, sustainability messages emphasizing collective efficacy and shared responsibility could strengthen employees’ green identity and participation in energy-saving or recycling programs. Likewise, emotion-based framing can enhance internal communication campaigns aimed at shifting organizational culture toward sustainability.

Thus, congruence between messages and consumers’ personal interests, activities, or values can increase their engagement, a fact that is also supported by previous studies showing that message specificity improves its reception by the audience ([39]; [90]).

Also, those corporate social responsibility (CSR) initiatives in which the message emphasizes consumer knowledge or interest in innovations are received more positively than general moral appeals; here, our results are in line with many recent studies showing that personalized messages lead to improved personal attitude, greater trust and, ultimately, better organizational outcomes ([1]; [2]; [63]). Our results are also congruent with other studies showing that it is necessary to personalize CSR communication and actions according to gender, as women often show stronger physiological reactions, while men report more pronounced cognitive effects ([11]; [26]).

Recognizing these differences can allow organizations to create more targeted messages that encourage pro-environmental behavior, but also take into account the emotional sensitivities and preferences identified through segmentation ([52]). Furthermore, we believe that the identification of seven distinct response clusters based on GSR provides a solid basis for future segmentation of consumer typologies, which will allow managers to develop communication strategies that tap into more diverse emotional and cognitive profiles.

Together, these results will lead to several possible managerial actions such as the following: designing communication campaigns that use segmentation based on physiological and cognitive response models; personalizing message content to match consumers’ gender and interests to maximize their engagement with organizations; and continuously monitoring and adapting campaigns based on audience biometric and behavioral feedback.

By implementing such strategies, organizations can improve their communication effectiveness, increase trust in CSR initiatives, and promote long-term pro-environmental behaviors among consumers. On the same basis, they can use message personalization to adjust their internal organizational behavior, optimizing information and decision-making processes so that the organization’s actions have a more precise and effective impact.

Limitations include the sample’s lack of cultural and demographic diversity, as well as the use of non-probabilistic sampling which restrict the generalizability of the findings. Given the gender imbalance in the sample, results related to gender differences should be interpreted with caution. Future studies should aim for a more balanced distribution to validate observed effects. Additionally, the controlled lab setting with static stimuli may not fully capture the dynamics of real-world media consumption. While GSR effectively measures physiological arousal, it does not capture emotional valence or cognitive processing. While this study offers relevant insights into emotional responses to sustainability messaging, several limitations should be acknowledged. First, the sample, although adequate for exploratory neuromarketing research, lacks demographic diversity and is relatively small, which limits the generalizability of the findings. The gender imbalance further suggests that gender-related effects should be interpreted cautiously. Second, the experimental design relied on static images presented in a controlled laboratory setting, which, although necessary for signal quality control, does not fully reflect real-world media environments in which individuals are typically exposed to dynamic multimedia content. Third, although galvanic skin response (GSR) provides a robust measure of physiological arousal, it does not capture emotional valence or cognitive processing. Reliance on a single biometric index therefore represents a constraint, as emotional engagement is multidimensional. Future research could address these limitations by recruiting more heterogeneous samples, incorporating additional biometric measures such as electroencephalography (EEG), heart rate variability (HRV), or eye-tracking, and employing more ecologically valid stimuli, including video formats and social media content. Such approaches may strengthen the validity and practical relevance of neuromarketing insights in sustainability communication.

Future research could build on the current cluster analysis by exploring how segment-level GSR response patterns relate to stable individual differences. Analyzing the distribution of cluster memberships within participants can reveal dominant physiological response styles, laying the foundation for data-driven personas and tailored communication strategies. Cognitively framed messages might resonate more with individuals showing ‘Cluster 1’ (Low Activation, Moderate Impact) patterns, while empowering, solution-oriented content could better engage those in ‘Cluster 3’ (High Peak Activation). A subject-level aggregation could be used for translating physiological response patterns into actionable audience segmentation strategies. To strengthen theoretical grounding, future studies could validate these response personas against established psychological constructs. Incorporating standardized measures (personality traits, environmental attitudes, cognitive processing styles, emotional regulation, and domain-specific knowledge) together with biometric data would improve our understanding of the psychological drivers behind physiological variability. Methodologically, expanding to multi-modal biometric approaches (e.g., facial expression analysis, HRV, EEG, eye tracking) and dynamic media formats (e.g., video, social media) would capture a broader range of emotional and cognitive responses. Cross-cultural and longitudinal designs could further assess the stability and external validity of response patterns.

Overall, this study advances the theoretical and applied understanding of emotional engagement with sustainability messages, emphasizing that effective communication must be nuanced, context-sensitive, and audience-aware, built upon empirical insights into emotional and physiological dynamics.

## Figures and Tables

**Figure 1 behavsci-15-01611-f001:**
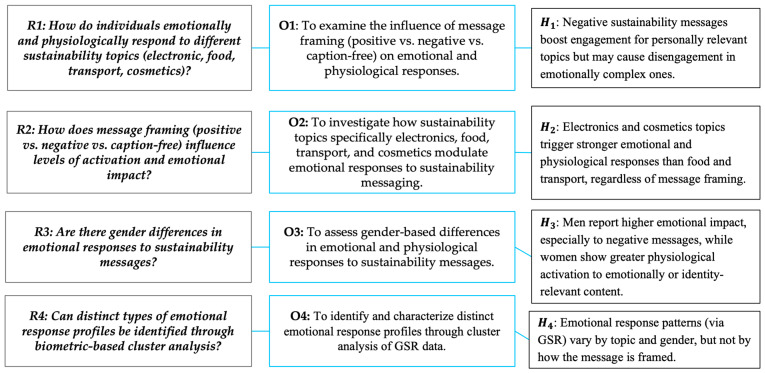
Structure of research design: from questions to hypotheses.

**Figure 2 behavsci-15-01611-f002:**
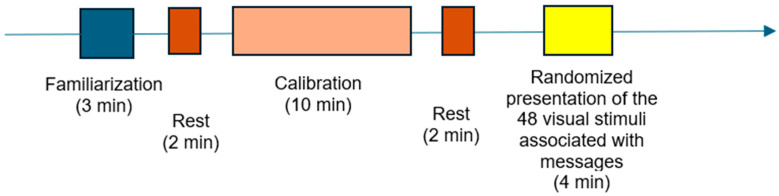
Sequential phases of the research protocol.

**Figure 3 behavsci-15-01611-f003:**
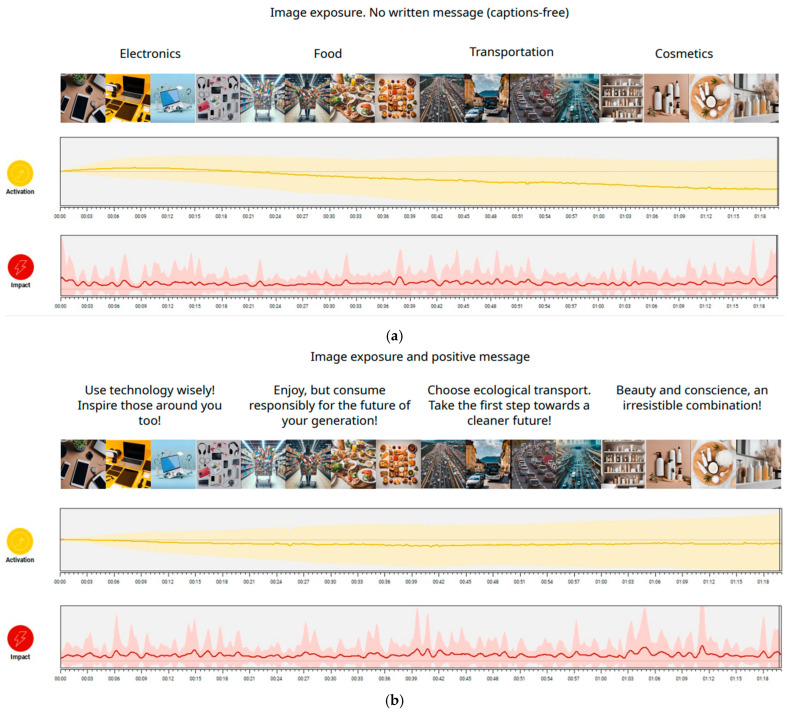
(**a**) Image exposure. No written message (captions-free). (**b**) Image exposure and positive message. (**c**) Image exposure and negative message.

**Figure 4 behavsci-15-01611-f004:**
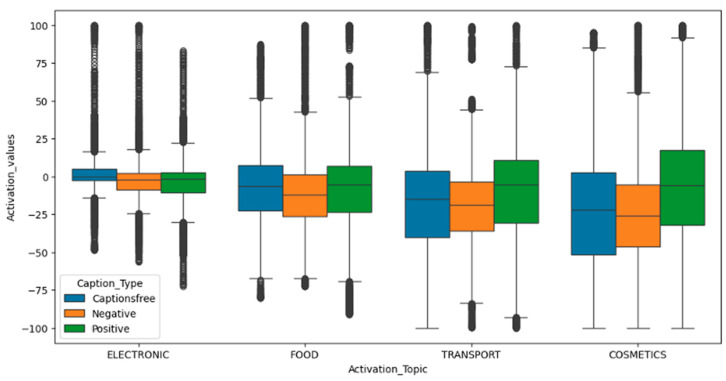
Boxplot of activation and impact responses by topic and caption type.

**Figure 5 behavsci-15-01611-f005:**
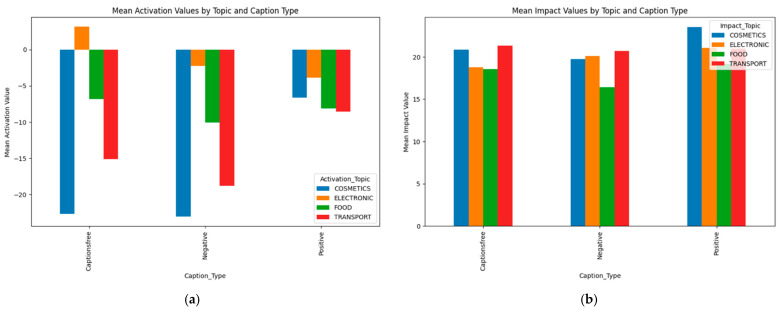
Comparison: (**a**) mean activation values by topic and caption type and (**b**) mean impact values by topic and caption type.

**Figure 6 behavsci-15-01611-f006:**
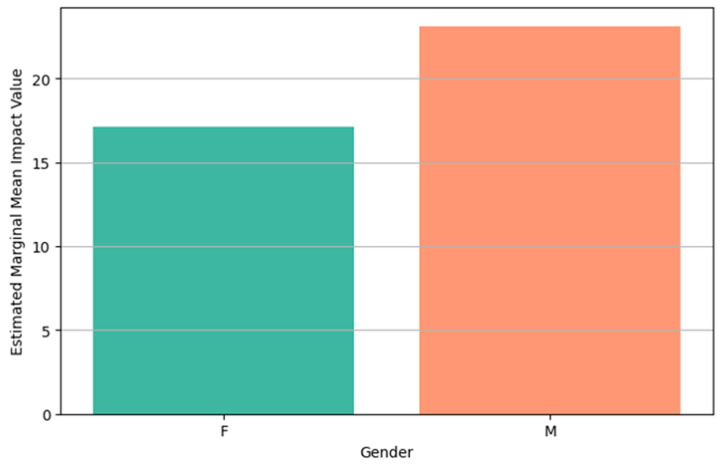
Main effect of gender on EMM impact values.

**Figure 7 behavsci-15-01611-f007:**
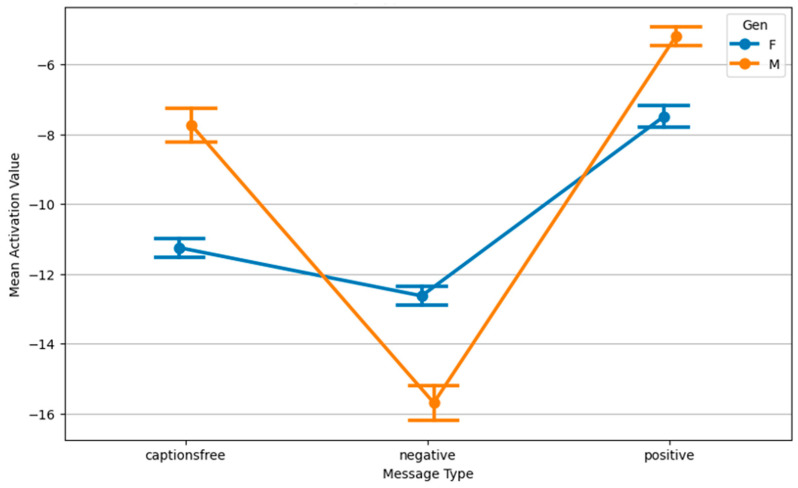
Interaction plot: message type and gender for activation value.

**Figure 8 behavsci-15-01611-f008:**
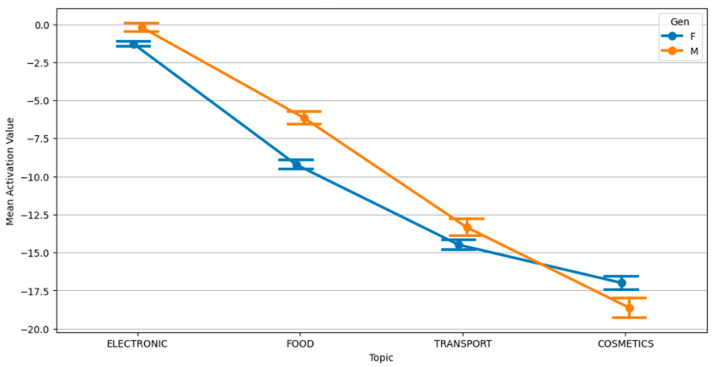
Interaction plot: topic and gender for activation value.

**Figure 9 behavsci-15-01611-f009:**
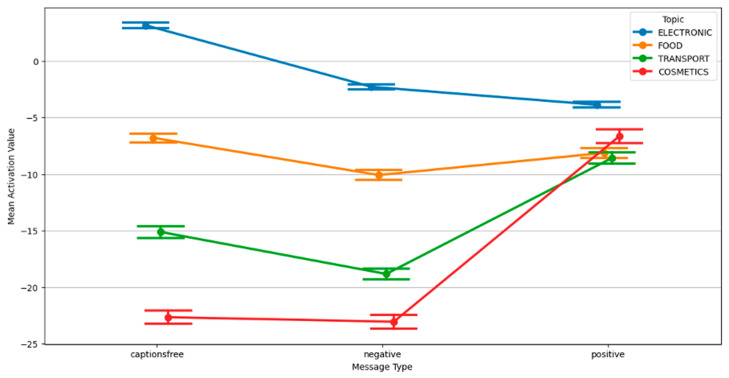
Interaction plot: message type and topic for activation value.

**Figure 10 behavsci-15-01611-f010:**
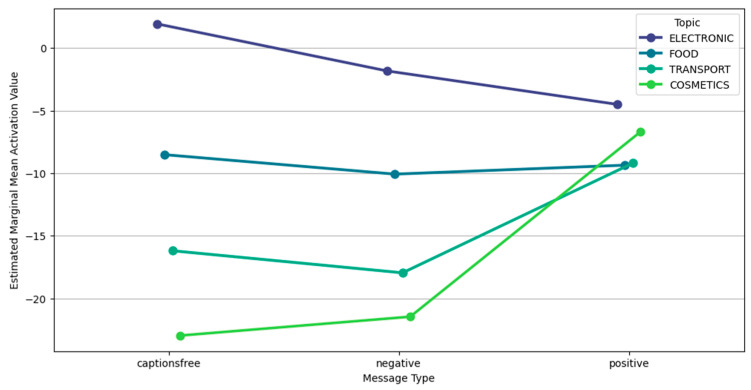
Interaction plot: message type and topic on EMM impact value (Gen = F).

**Figure 11 behavsci-15-01611-f011:**
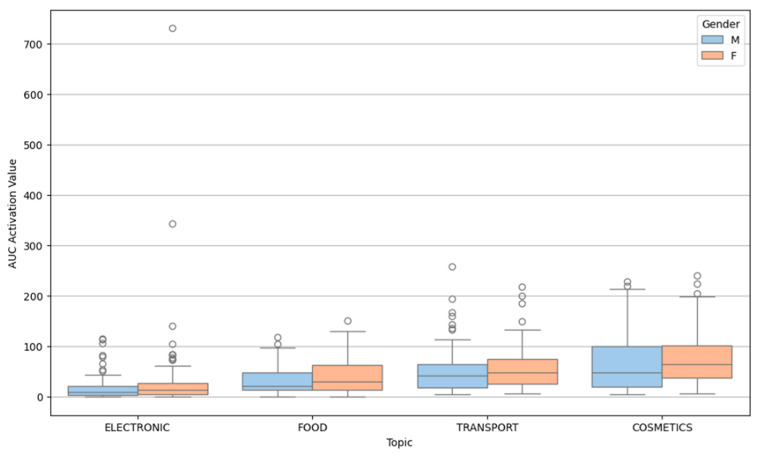
Box plot of AUC activation value by topic and gender.

**Figure 12 behavsci-15-01611-f012:**
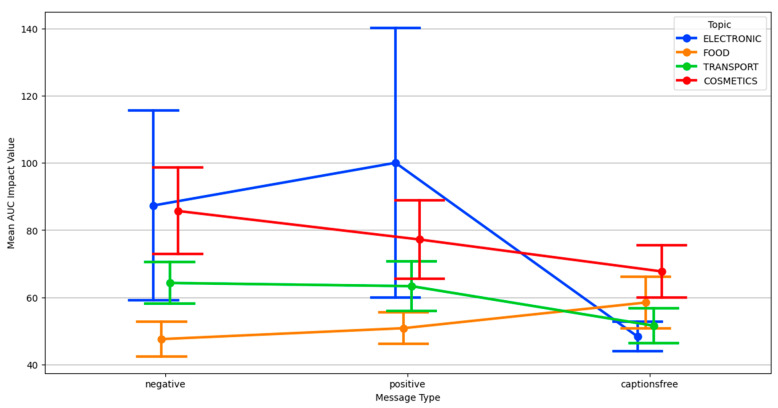
Interaction plot of message type and topic for AUC impact value.

**Figure 13 behavsci-15-01611-f013:**
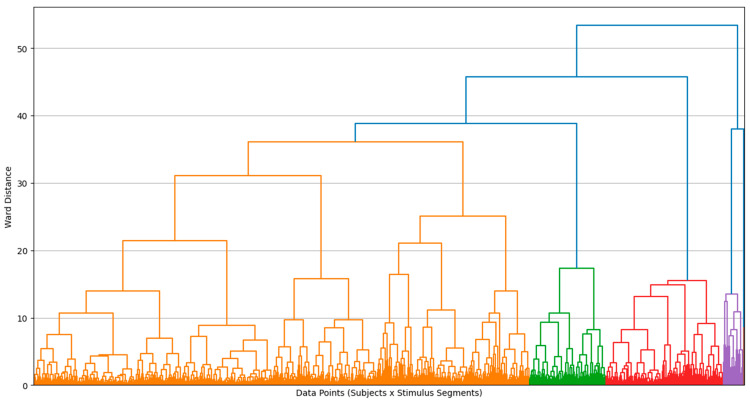
Hierarchical clustering dendrogram showing seven GSR-based emotional response patterns across all stimulus segments. The colors differentiate the clusters.

**Figure 14 behavsci-15-01611-f014:**
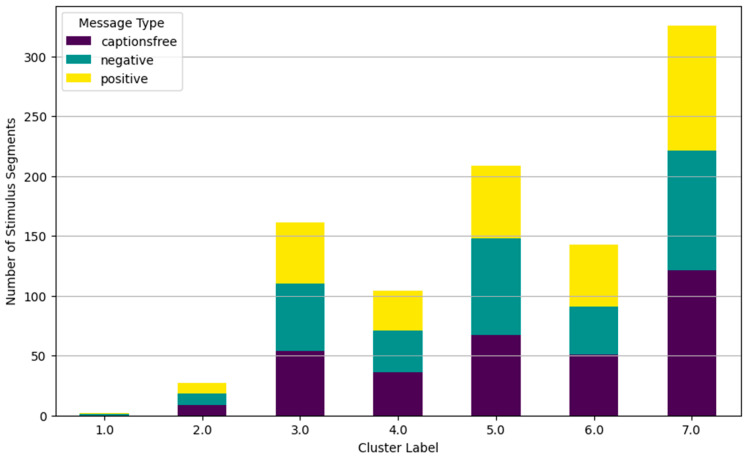
Cluster distribution by message type (hierarchical clustering—7 clusters).

**Figure 15 behavsci-15-01611-f015:**
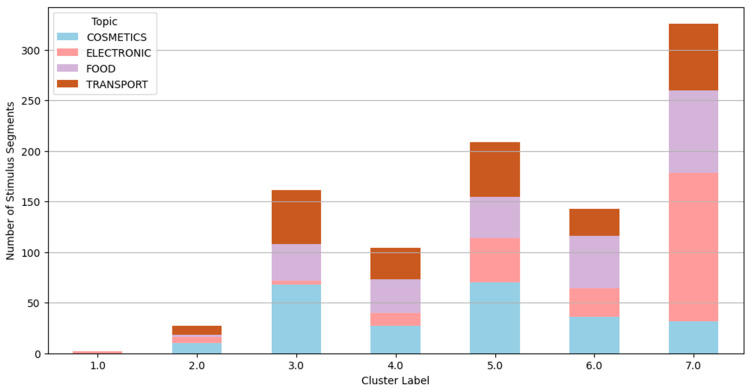
Cluster distribution by topic (hierarchical clustering—7 clusters).

**Figure 16 behavsci-15-01611-f016:**
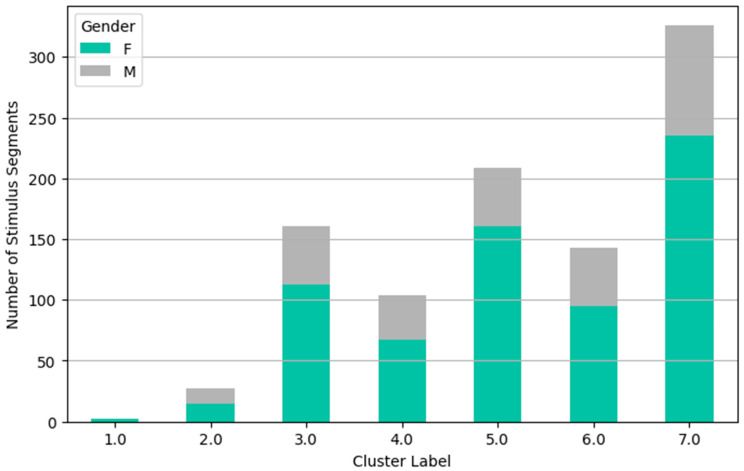
Cluster distribution by gender (hierarchical clustering—7 clusters).

**Table 1 behavsci-15-01611-t001:** The results of cluster membership.

Cluster vs. Message Type	Although the distribution of message types (captions-free, negative, positive) across clusters was inspected, the chi-squared test was not significant (*p* = 0.829). This indicates that message framing alone does not strongly predict cluster membership.
Cluster vs. Topic	For example, Cluster 3 is predominantly associated with electronic stimuli, Cluster 2 with food, and Cluster 1 with transport and cosmetics. The chi-squared test was highly significant (*p* = 0.000), underscoring that the topic of the stimulus is a major determinant of GSR response patterns.
Cluster vs. Gender	The distribution of gender across clusters, displayed slight statistically significant differences (*p* = 0.076). Although females are numerically dominant across clusters, certain clusters (Clusters 2 and 3) exhibit a relatively higher proportion of male responses. This weaker association suggests that gender modulates the likelihood of exhibiting particular response patterns.

## Data Availability

Available data on request.

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
