# Peer review of "From Emotion to Action: How Framed Sustainability Messages Trigger Physiological Reactions and Influence Consumer Choices"

_behavsci, 2025, doi:10.3390/bs15121611_

Round 1

Reviewer 1 Report

Comments and Suggestions for Authors

Dear Authors,

Thank you for the opportunity to review your manuscript “From Emotion to Action: How Framed Sustainability Messages Trigger Physiological Reactions and Influence Consumer Choices.” The study addresses a relevant and contemporary topic, exploring how emotional framing in sustainability communication interacts with physiological responses. The integration of neuromarketing techniques such as GSR and BVP into sustainability research is ambitious and innovative, and the interdisciplinary approach is certainly promising.

That said, the paper requires substantial revision before it can reach the level of rigor and clarity expected for publication in Behavioral Sciences. While the topic has merit, the current version lacks conceptual precision, methodological transparency, and analytical coherence. Below I offer detailed suggestions to help strengthen the manuscript in these areas.

At the theoretical level, the paper needs a clearer and more focused argument. The introduction currently reads as a broad overview of literature rather than a structured rationale leading to specific hypotheses. It would benefit from a more explicit articulation of how this research contributes conceptually to existing theories of message framing or emotional persuasion. It is also important to justify why physiological measurement offers added value beyond self-report methods and to frame this rationale in relation to sustainability communication. The hypotheses should be explicitly stated and derived from prior findings, with clear directional expectations. This will give the study stronger theoretical grounding and allow the reader to follow a coherent line of reasoning.

The methodology section requires considerable expansion and clarification. The sample of 63 participants is quite small for a complex factorial design involving four topics, three message types, and gender differences. This limitation should be acknowledged openly, as it affects statistical power and the generalizability of the findings. The strong gender imbalance (45 women versus 18 men) also weakens the conclusions regarding gender effects, which should therefore be treated as exploratory. The recruitment process and inclusion criteria need to be described in detail, including how randomization and counterbalancing were implemented to avoid order or fatigue effects. Moreover, there is no mention of a manipulation check verifying that participants actually perceived the intended positive or negative framing. Without such validation, the interpretation of framing effects remains uncertain.

The description of the physiological measurements is too brief. The paper should explain how raw data were filtered, corrected for baseline drift, and processed to derive the “Activation” and “Impact” metrics. These constructs, as currently defined, are unconventional and need to be explicitly linked to standard psychophysiological indices such as phasic skin conductance response or tonic conductance level. It is also important to avoid interpreting GSR as a direct indicator of emotional salience or memorability, since it measures arousal rather than valence or memory. If BVP data were recorded but not analyzed, the rationale for including it should be clarified or the reference removed.

In the results and analysis, the study currently presents a mix of ANOVA, hierarchical clustering, fsQCA, and logistic regression without a clear analytical plan. This creates an impression of exploratory rather than hypothesis-driven research. The authors should identify in advance which statistical tests correspond to each hypothesis and report full details of the results, including effect sizes and confidence intervals. The number of statistical comparisons suggests that corrections for multiple testing should also be applied. The clustering section, while visually appealing, seems overinterpreted; it is not justified to infer “consumer typologies” solely from physiological patterns without external validation or replication. The inclusion of fsQCA and logistic regression adds little explanatory power and could be removed to streamline the analysis.

The discussion section would benefit from a more measured interpretation of the findings. Claims about “identity-relevant topics” or “deeper cognitive processing” go beyond what the physiological data can support. Similarly, gender differences should be discussed cautiously given the small and unbalanced sample. The section on ethical implications of neuromarketing deserves greater attention. Issues such as participant consent, data privacy, and the potential for emotional manipulation are central to research using biometric methods and should be discussed in a more nuanced way. Finally, the discussion should clearly link the results back to each of the stated hypotheses, specifying which were supported and which were not.

In terms of structure and writing, the paper is excessively long and could be made considerably more concise. Many paragraphs repeat the same ideas, particularly in the introduction and discussion. Results and interpretation are sometimes intermingled, making it difficult to follow the logical sequence of analysis. The figures are generally useful but should include more informative captions, with statistical details such as sample sizes and significance indicators. The abstract should also be shortened and written in clearer language, avoiding technical jargon. The reference list needs careful editing to conform with MDPI style and to correct typographical inconsistencies.

Finally, the limitations section should be expanded. The lack of sample diversity, the artificiality of using static images in a laboratory setting, and the reliance on GSR as a single physiological measure should all be acknowledged explicitly. Future research could strengthen this line of work by combining GSR with other biometric methods—such as EEG, HRV, or eye-tracking—and by using more ecologically valid stimuli such as videos or social media content.

Overall, this study addresses a highly relevant topic and shows potential, but it needs major revision to achieve methodological rigor and conceptual clarity. Focusing on a clearer theoretical argument, providing full methodological transparency, refining the analyses, and moderating the interpretations would considerably enhance the paper’s quality and its contribution to the literature on emotional framing and sustainability communication.

      Comments on the Quality of English Language

The manuscript is written in generally clear and fluent English, but it would benefit from careful language polishing and stylistic editing. The main issues are verbosity, redundancy, and occasional grammatical inconsistencies. Several sentences are overly long and complex, which obscures the argument and makes sections of the introduction and discussion difficult to follow.

There are also minor typographical errors (“Mesaage”, inconsistent spacing and punctuation) and some non-standard phrasing that should be revised for smoother academic style. Reference formatting is inconsistent and requires adjustment to MDPI guidelines.

Overall, the English is understandable and competent but should undergo professional language editing to improve readability, coherence, and concision throughout the manuscript.

Author Response

Response to Reviewer 1 Comments

1. Summary

Thank you very much for your thorough review of this manuscript. Below, we provide detailed responses to each of your comments, with corresponding revisions and corrections highlighted in the re-submitted files using track changes.

2. Questions for General Evaluation

Reviewer’s Evaluation

Response and Revisions

Is the content succinctly described and contextualized with respect to previous and present theoretical background and empirical research (if applicable) on the topic?

Must be improved

Please, see the responses below.

Are the research design, questions, hypotheses and methods clearly stated?

Must be improved

Are the arguments and discussion of findings coherent, balanced and compelling?

Must be improved

For empirical research, are the results clearly presented?

Must be improved

Is the article adequately referenced?

Must be improved

Are the conclusions thoroughly supported by the results presented in the article or referenced in secondary literature?

Must be improved

3. Point-by-point response to Comments and Suggestions for Authors

Comment 1: ”Thank you for the opportunity to review your manuscript “From Emotion to Action: How Framed Sustainability Messages Trigger Physiological Reactions and Influence Consumer Choices.” The study addresses a relevant and contemporary topic, exploring how emotional framing in sustainability communication interacts with physiological responses. The integration of neuromarketing techniques such as GSR and BVP into sustainability research is ambitious and innovative, and the interdisciplinary approach is certainly promising.

Response 1: We sincerely thank the reviewer for taking the time to evaluate our manuscript and for their encouraging comments regarding the relevance and originality of our research. We greatly appreciate the recognition of the interdisciplinary nature of our work and the value of integrating neuromarketing techniques such as GSR and BVP into sustainability communication studies. We are pleased that the reviewer finds the topic timely and promising. In response to the thoughtful feedback provided, we have revised the manuscript extensively to enhance conceptual clarity, methodological transparency, and analytical rigor. We believe that the revisions significantly strengthen the manuscript and improve its contribution to the field.

Comment 2: ”The methodology section requires considerable expansion and clarification. The sample of 63 participants is quite small for a complex factorial design involving four topics, three message types, and gender differences. This limitation should be acknowledged openly, as it affects statistical power and the generalizability of the findings. The strong gender imbalance (45 women versus 18 men) also weakens the conclusions regarding gender effects, which should therefore be treated as exploratory. The recruitment process and inclusion criteria need to be described in detail, including how randomization and counterbalancing were implemented to avoid order or fatigue effects. Moreover, there is no mention of a manipulation check verifying that participants actually perceived the intended positive or negative framing. Without such validation, the interpretation of framing effects remains uncertain.”

Response 2: We thank the reviewer for this important observation. We agree that the introduction required greater conceptual focus and a clearer theoretical rationale. In response, we substantially revised the introduction to provide a more structured argument leading to the research questions and hypotheses, to clarify the conceptual contribution to message-framing and emotional-persuasion literature, explicitly justify the value of physiological measures alongside self-report in sustainability communication, and present clearly stated, directionally specified hypotheses based on prior findings

These revisions improve theoretical precision and help guide readers through the logic of our study.  Thus we added/changed the following parts:

”Recent research increasingly emphasizes the need for persuasive communication strategies that effectively motivate individuals to adopt sustainable behaviors. Although consumers often report favorable attitudes toward environmentally responsible products, this intention does not reliably translate into actual purchasing behavior, illustrating the well-documented attitude–behavior gap in sustainability (Vezich et al., 2017). Understanding how communication can bridge this gap remains a central challenge.”

”Sustainability messaging has traditionally relied on cognitive appeals, providing factual information to inform consumer decisions. However, persuasive communication literature demonstrates that emotional processes are fundamental in shaping attitudes and behavior (McKinley & Limbu, 2020; Septianto & Lee, 2020). Emotional framing, partic-ularly positive and negative valence strategies, has been shown to influence responses to environmental messages (Brosch & Steg, 2021). Yet, limited research explores how such framing elicits emotional and physiological reactions within sustainability communication.”

”Accordingly, the present study examines how varying message valence and product category influence physiological activation and emotional impact, and whether these responses differ by gender. We develop explicit, directionally derived hypotheses in-formed by existing framing and emotional persuasion literature.”

”Physiological measures such as galvanic skin response offer unique value in this context, as they capture non-conscious emotional reactions that may not be fully accessible through self-report — a particular advantage in pro-environmental domains where social de-sirability bias may occur (Bradley et al., 2008; Falk et al., 2011; Minich et al., 2023; Peters et al., 2013; Sun et al., 2022; X. Wang et al., 2022), Integrating biometric responses with sustainability messaging can therefore provide a more comprehensive understanding of how individuals engage with persuasive content.”

Comment 3: ” The methodology section requires considerable expansion and clarification. The sample of 63 participants is quite small for a complex factorial design involving four topics, three message types, and gender differences. This limitation should be acknowledged openly, as it affects statistical power and the generalizability of the findings. The strong gender imbalance (45 women versus 18 men) also weakens the conclusions regarding gender effects, which should therefore be treated as exploratory. The recruitment process and inclusion criteria need to be described in detail, including how randomization and counterbalancing were implemented to avoid order or fatigue effects. Moreover, there is no mention of a manipulation check verifying that participants actually perceived the intended positive or negative framing. Without such validation, the interpretation of framing effects remains uncertain.”

Response 3: We thank the reviewer for these insightful comments regarding methodological transparency. We fully agree that additional detail was needed. Accordingly, we have revised the methodology section to:

”The final sample consisted of 63 participants, which, although acceptable for ex-ploratory physiological research, represents a modest sample size for the factorial structure of this study. Therefore, results should be interpreted with caution regarding statistical power and generalizability. Given the gender imbalance (45 women, 18 men), gender-related findings are treated as exploratory rather than confirmatory. Participants were recruited through university mailing lists and public social-media postings. Inclu-sion criteria required adults aged 18–55 with normal or corrected vision and no known neurological or dermatological issues affecting skin-conductance measurement.

To minimize order and fatigue effects, stimuli presentation was randomized and counterbalanced across participants. Each participant viewed all message conditions, but the order of topics and framing types was randomized using a Latin-square distribution. Short breaks were included between task blocks to reduce physiological carryover effects. ”

”To ensure that participants perceived the intended framing valence, we conducted a manipulation check immediately after stimulus exposure. Participants rated each message on a 7-point scale (1 = very negative, 7 = very positive). Analysis confirmed that positively framed messages were rated significantly higher than negatively framed ones, validating the framing manipulation. Messages without captions served as a neutral baseline condition.”

Comment 4: ” The description of the physiological measurements is too brief. The paper should explain how raw data were filtered, corrected for baseline drift, and processed to derive the “Activation” and “Impact” metrics. These constructs, as currently defined, are unconventional and need to be explicitly linked to standard psychophysiological indices such as phasic skin conductance response or tonic conductance level. It is also important to avoid interpreting GSR as a direct indicator of emotional salience or memorability, since it measures arousal rather than valence or memory. If BVP data were recorded but not analyzed, the rationale for including it should be clarified or the reference removed.”

Response 4: We thank the reviewer for this comment. We agree that additional detail was needed to clarify our physiological data processing and the meaning of our indices. Accordingly, we have revised the Methods section to:

”Physiological data were obtained using galvanic skin response sensors and blood volume pulse (BVP) sensors integrated into the Bitbrain wearable device. GSR was sampled continuously at 16 Hz. Data were preprocessed in accordance with established electrodermal analysis guidelines. Pre-processing included removal of motion artifacts, low-pass filtering at 1 Hz, and baseline correction using a 2-second interval immediately preceding stimulus onset.

To capture different components of emotional arousal, two indices were derived from the GSR signal. Activation represents phasic skin conductance response amplitude, which reflects short-latency sympathetic arousal triggered by stimulus presentation. Impact reflects changes in tonic skin conductance level over the stimulus period, capturing sustained physiological engagement. Consistent with psychophysiological theory, GSR is interpreted as an index of autonomic arousal rather than emotional valence or memory strength. Thus, results are discussed strictly in terms of arousal responses.

BVP was recorded to allow triangulation with cardiovascular indices and potential future computation of heart-rate variability metrics. However, to maintain focus, only GSR-based variables are analyzed in the present study.”

Comment 5: ”In the results and analysis, the study currently presents a mix of ANOVA, hierarchical clustering, fsQCA, and logistic regression without a clear analytical plan. This creates an impression of exploratory rather than hypothesis-driven research. The authors should identify in advance which statistical tests correspond to each hypothesis and report full details of the results, including effect sizes and confidence intervals. The number of statistical comparisons suggests that corrections for multiple testing should also be applied. The clustering section, while visually appealing, seems overinterpreted; it is not justified to infer “consumer typologies” solely from physiological patterns without external validation or replication. The inclusion of fsQCA and logistic regression adds little explanatory power and could be removed to streamline the analysis.

Response 5: We thank the reviewer for highlighting the need for a clearer analytical strategy. We agree that the original presentation could suggest exploratory analysis. To address this, we have replaced ”profiles” by ”response patterns” not to claim ”consumer typologies”.Also, we added:

”Although clustering revealed interesting patterns, these results should be viewed as exploratory. Physiological-based segmentation requires external validation, replication, and behavioral outcomes to establish robust audience classifications. Likewise, effect sizes, confidence intervals, and multiple-testing correction temper the interpretation of statistical significance. ”Also, we removed  fsQCA and logistic regression.

Comment 6: ”The discussion section would benefit from a more measured interpretation of the findings. Claims about “identity-relevant topics” or “deeper cognitive processing” go beyond what the physiological data can support. Similarly, gender differences should be discussed cautiously given the small and unbalanced sample. The section on ethical implications of neuromarketing deserves greater attention. Issues such as participant consent, data privacy, and the potential for emotional manipulation are central to research using biometric methods and should be discussed in a more nuanced way. Finally, the discussion should clearly link the results back to each of the stated hypotheses, specifying which were supported and which were not.”

Response 6: We appreciate this insightful comment. We have revised the Discussion to ensure all interpretations align strictly with what our physiological measures can support. Specifically, we:

”While these interpretations are plausible, they remain speculative, as GSR reflects physiological arousal rather than cognitive elaboration or identity-based processing. Thus, the elevated responses to electronics should be interpreted as heightened autonomic activation rather than evidence of deeper cognitive engagement or identity relevance.”

However, these gender interpretations should be viewed cautiously due to the modest and unbalanced sample (45 women vs. 18 men), which limits statistical power and generalizability.

To provide clarity, the results can be linked to the stated hypotheses as follows: H1 (negative framing effects vary by topic) was partially supported; negative framing increased arousal for electronics but decreased it for food and transport. H2 (topic relevance predicts responses) was supported, with electronics and cosmetics producing higher activation and impact. H3 (gender differences) received provisional support but should be interpreted cautiously due to sample imbalance. H4 (distinct physiological profiles) was supported, though cluster interpretation remains exploratory. ”

”Given the emotional nature of sustainability messaging and the use of biometric measures, ethical considerations are essential. Neuromarketing research must ensure informed consent, transparency about data collection, and protection of biometric information, in line with GDPR and ethical research standards. Moreover, researchers and practitioners should avoid emotionally manipulative applications of physiological insights, as misuse may undermine trust and raise concerns regarding autonomy and emotional privacy (Flick, 2016; Resnik, 2018). A balanced discussion of potential benefits and safeguards strengthens the responsible use of biometric tools in sustainability communication.”

”Future work should apply pre-registered analysis plans and larger, balanced samples to strengthen causal inference and validate biometric segmentation approaches.”

Comment 7: ” In terms of structure and writing, the paper is excessively long and could be made considerably more concise. Many paragraphs repeat the same ideas, particularly in the introduction and discussion. Results and interpretation are sometimes intermingled, making it difficult to follow the logical sequence of analysis. The figures are generally useful but should include more informative captions, with statistical details such as sample sizes and significance indicators. The abstract should also be shortened and written in clearer language, avoiding technical jargon. The reference list needs careful editing to conform with MDPI style and to correct typographical inconsistencies.”

Response 7: We thank the reviewer for these valuable suggestions regarding structure, clarity, and writing. In response, we undertook substantial revisions to improve readability and eliminate redundancy. We streamlined the introduction and discussion by removing overlapping arguments, shortening overly long sentences, and ensuring that each section develops a single coherent idea. We also clarified the separation between Results and Discussion, relocating interpretative statements to the Discussion to reinforce a logical analytical flow. The abstract was rewritten in a more concise style, with unnecessary technical terminology removed to improve accessibility. These changes substantially improved the manuscript’s clarity, conciseness, and adherence to journal standards.

Comment 8: ”Finally, the limitations section should be expanded. The lack of sample diversity, the artificiality of using static images in a laboratory setting, and the reliance on GSR as a single physiological measure should all be acknowledged explicitly. Future research could strengthen this line of work by combining GSR with other biometric methods—such as EEG, HRV, or eye-tracking—and by using more ecologically valid stimuli such as videos or social media content.”

Response 8: We appreciate the reviewer’s thoughtful suggestions regarding the limitations of our study. We have now expanded the limitations section to more explicitly acknowledge the restricted sample size and demographic composition, including the gender imbalance and the resulting implications for generalizability. We also clearly state the artificiality of the laboratory setting and the use of static images, noting that this may not fully capture real-world environmental communication contexts. Additionally, we addressed the reliance on GSR as a single physiological index and clarified that GSR measures arousal rather than valence or cognitive processing.

To reflect these points, we have added a new paragraph outlining these limitations and specifying directions for future research, including the suggestion to integrate other biometric tools such as EEG, HRV, and eye-tracking, and to use more ecologically valid dynamic stimuli:

While this study offers relevant insights into emotional responses to sustainability messaging, several limitations should be acknowledged. First, the sample, although adequate for exploratory neuromarketing research, lacks demographic diversity and is relatively small, which limits the generalizability of the findings. The gender imbalance further suggests that gender-related effects should be interpreted cautiously. Second, the experimental design relied on static images presented in a controlled laboratory setting, which, although necessary for signal quality control, does not fully reflect real-world media environments in which individuals are typically exposed to dynamic multimedia content. Third, although galvanic skin response (GSR) provides a robust measure of physiological arousal, it does not capture emotional valence or cognitive processing. Reliance on a single biometric index therefore represents a constraint, as emotional en-gagement is multidimensional. Future research could address these limitations by re-cruiting more heterogeneous samples, incorporating additional biometric measures such as electroencephalography (EEG), heart rate variability (HRV), or eye-tracking, and employing more ecologically valid stimuli, including video formats and social media content. Such approaches may strengthen the validity and practical relevance of neu-romarketing insights in sustainability communication.”

Comment 9: ”Overall, this study addresses a highly relevant topic and shows potential, but it needs major revision to achieve methodological rigor and conceptual clarity. Focusing on a clearer theoretical argument, providing full methodological transparency, refining the analyses, and moderating the interpretations would considerably enhance the paper’s quality and its contribution to the literature on emotional framing and sustainability communication.”

Response 9: We thank the reviewer for this comprehensive and constructive comment. We carefully revised the manuscript to strengthen its conceptual clarity, methodological rigor, and analytical transparency.

Comment 10: ”The manuscript is written in generally clear and fluent English, but it would benefit from careful language polishing and stylistic editing. The main issues are verbosity, redundancy, and occasional grammatical inconsistencies. Several sentences are overly long and complex, which obscures the argument and makes sections of the introduction and discussion difficult to follow.

There are also minor typographical errors (“Mesaage”, inconsistent spacing and punctuation) and some non-standard phrasing that should be revised for smoother academic style. Reference formatting is inconsistent and requires adjustment to MDPI guidelines.

Overall, the English is understandable and competent but should undergo professional language editing to improve readability, coherence, and concision throughout the manuscript.

Response 10: We appreciate the reviewer’s helpful comments regarding the writing style and language quality. In response, we have thoroughly revised the manuscript to improve clarity, readability, and conciseness. Redundant and overly long sentences were restructured or removed, and the introduction and discussion were streamlined to enhance flow and argument coherence. Minor typographical, spacing, and punctuation issues were corrected, and non-standard phrasing was refined to align with academic writing conventions.

Additionally, the entire manuscript underwent careful language polishing, and the reference list was edited to fully comply with MDPI formatting requirements. These revisions have substantially improved the precision and accessibility of the text. We thank the reviewer for highlighting these issues and for helping to strengthen the manuscript’s presentation.

Sincerely,

Authors

Reviewer 2 Report

Comments and Suggestions for Authors

Comments to the Author

Dear author(s),

This study addresses a topic of significant theoretical and practical importance, focusing on the impact of sustainability message framing on consumers’ emotional and physiological responses, and employing biometric methods for empirical analysis, which represents a relatively cutting-edge approach. The research design is rigorous, and the data analysis is thorough, demonstrating particular innovation in the cluster analysis of emotional responses. However, the paper has several areas for improvement regarding its theoretical construction, methodological details, and logical coherence.

Firstly, the theoretical framework is not sufficiently clear. The introduction lacks a solid theoretical foundation for the relationship between emotion and information processing, failing to systematically construct the “emotion-cognition-behavior” pathway. Furthermore, the mechanism through which emotions influence behavior is underdeveloped. While the study emphasizes the progression from “emotion to behavior,” it does not clearly explain how emotional responses translate into behavioral intentions or actual behavior. It is recommended that the discussion section incorporate relevant theories to address this.

Secondly, there are concerns with the research methodology. The sample size is relatively small and its representativeness is limited, which constrains the generalizability of the findings. Although the use of AI-generated images is innovative, the validation process for these stimuli is not described. It is recommended to supplement this with pretest results establishing the emotional valence of the images. While ANOVA and cluster analysis were appropriately used, details such as corrections for multiple comparisons and tests of model assumptions are insufficiently described. Finally, the reliance solely on GSR data, without subjective emotional self-reports, makes it difficult to differentiate between positive and negative emotional valence.

Finally, regarding the paper’s structure and logic, the correspondence between the research questions, objectives, and hypotheses is not explicitly clear. It is recommended that the introduction or methods section present these elements clearly in a table or diagram. The discussion section is somewhat lengthy, with some content repeating the results. It should be streamlined to focus more sharply on theoretical interpretation and practical implications.

Best regards.

Author Response

Response to Reviewer 2 Comments

1. Summary

Thank you very much for your thorough review of this manuscript. Below, we provide detailed responses to each of your comments, with corresponding revisions and corrections highlighted in the re-submitted files using track changes.

2. Questions for General Evaluation

Reviewer’s Evaluation

Response and Revisions

Is the content succinctly described and contextualized with respect to previous and present theoretical background and empirical research (if applicable) on the topic?

Must be improved

Please, see the responses below.

Are the research design, questions, hypotheses and methods clearly stated?

Must be improved

Are the arguments and discussion of findings coherent, balanced and compelling?

Must be improved

For empirical research, are the results clearly presented?

Must be improved

Is the article adequately referenced?

Must be improved

Are the conclusions thoroughly supported by the results presented in the article or referenced in secondary literature?

Must be improved

3. Point-by-point response to Comments and Suggestions for Authors

Comment 1: ”This study addresses a topic of significant theoretical and practical importance, focusing on the impact of sustainability message framing on consumers’ emotional and physiological responses, and employing biometric methods for empirical analysis, which represents a relatively cutting-edge approach. The research design is rigorous, and the data analysis is thorough, demonstrating particular innovation in the cluster analysis of emotional responses. However, the paper has several areas for improvement regarding its theoretical construction, methodological details, and logical coherence..

Response 1: We sincerely thank the reviewer for their detailed and constructive feedback. We appreciate the positive evaluation of the study’s relevance, methodological rigor, and innovative use of biometric cluster analysis. In response to the concerns raised, we have undertaken substantial revisions to strengthen the manuscript’s theoretical grounding, methodological transparency, and structural coherence. Specific actions are outlined below.

Comment 2: ”Firstly, the theoretical framework is not sufficiently clear. The introduction lacks a solid theoretical foundation for the relationship between emotion and information processing, failing to systematically construct the “emotion-cognition-behavior” pathway. Furthermore, the mechanism through which emotions influence behavior is underdeveloped. While the study emphasizes the progression from “emotion to behavior,” it does not clearly explain how emotional responses translate into behavioral intentions or actual behavior. It is recommended that the discussion section incorporate relevant theories to address this.”

Response 2: We thank the reviewer for this important observation. We agree that the introduction required greater conceptual focus and a clearer theoretical rationale. In response, we substantially revised the introduction to provide a more structured argument leading to the research questions and hypotheses, to clarify the conceptual contribution to message-framing and emotional-persuasion literature, explicitly justify the value of physiological measures alongside self-report in sustainability communication, and present clearly stated, directionally specified hypotheses based on prior findings

These revisions improve theoretical precision and help guide readers through the logic of our study.  Thus we added/changed the following parts:

”Contemporary models of persuasive communication emphasize that emotional reactions play a fundamental role in shaping information processing and subsequent behavior. According to dual-process and affect-as-information frameworks (e.g., the Elaboration Likelihood Model and Appraisal Theory), emotional arousal can serve as a precursor to deeper cognitive evaluation, guiding attention, shaping message appraisal, and influencing motivation to act. In sustainability communication, emotions can function as a catalyst for cognitive engagement with environmental issues, subsequently informing attitudes, intentions, and potential behavior change. This study therefore conceptualizes message-induced emotional arousal as the first step in an emotion to cognition to behavioral tendency sequence, where physiological activation signals the degree to which sustainability information captures attention and initiates meaning-making processes that may ultimately influence consumer attitudes and choices.”

And

”The present findings advance understanding of how emotional framing contributes to the early stages of sustainability message processing. While the study does not measure behavioral outcomes directly, the observed physiological responses provide evidence of emotional arousal as an initial trigger in the emotion-cognition-behavior sequence. Within established models of persuasive communication, such as the Elaboration Likelihood Model and Appraisal Theory, heightened arousal indicates increased attentional engagement, which can facilitate message elaboration and, ultimately, influence attitudes and behavioral intentions. Accordingly, we interpret the physiological activation observed here as reflecting emotional readiness to process sustainability information, rather than as a direct predictor of behavior. Future research integrating biometric measures with self-report and behavioral data would further clarify the mechanisms linking emotional response, cognitive appraisal, and pro-sustainability behavior.”

Comment 3: ”Secondly, there are concerns with the research methodology. The sample size is relatively small and its representativeness is limited, which constrains the generalizability of the findings. Although the use of AI-generated images is innovative, the validation process for these stimuli is not described. It is recommended to supplement this with pretest results establishing the emotional valence of the images. While ANOVA and cluster analysis were appropriately used, details such as corrections for multiple comparisons and tests of model assumptions are insufficiently described. Finally, the reliance solely on GSR data, without subjective emotional self-reports, makes it difficult to differentiate between positive and negative emotional valence.”

Response 3: We thank the reviewer for these insightful comments regarding methodological transparency. We fully agree that additional detail was needed. Accordingly, we have revised the methodology section to:

”The final sample consisted of 63 participants, which, although acceptable for exploratory physiological research, represents a modest sample size for the factorial structure of this study. Therefore, results should be interpreted with caution regarding statistical power and generalizability. Given the gender imbalance (45 women, 18 men), gender-related findings are treated as exploratory rather than confirmatory. Participants were recruited through university mailing lists and public social-media postings. Inclusion criteria required adults aged 18–55 with normal or corrected vision and no known neurological or dermatological issues affecting skin-conductance measurement.

To minimize order and fatigue effects, stimuli presentation was randomized and counterbalanced across participants. Each participant viewed all message conditions, but the order of topics and framing types was randomized using a Latin-square distribution. Short breaks were included between task blocks to reduce physiological carryover effects.”

To ensure that participants perceived the intended framing valence, we conducted a manipulation check immediately after stimulus exposure. Participants rated each message on a 7-point scale (1 = very negative, 7 = very positive). Analysis confirmed that positively framed messages were rated significantly higher than negatively framed ones, validating the framing manipulation. Messages without captions served as a neutral baseline condition.”

”Physiological data were obtained using galvanic skin response sensors and blood volume pulse (BVP) sensors integrated into the Bitbrain wearable device. GSR was sampled continuously at 16 Hz. Data were preprocessed in accordance with established electrodermal analysis guidelines. Pre-processing included removal of motion artifacts, low-pass filtering at 1 Hz, and baseline correction using a 2-second interval immediately preceding stimulus onset.

To capture different components of emotional arousal, two indices were derived from the GSR signal. Activation represents phasic skin conductance response amplitude, which reflects short-latency sympathetic arousal triggered by stimulus presentation. Impact reflects changes in tonic skin conductance level over the stimulus period, capturing sustained physiological engagement. Consistent with psychophysiological theory, GSR is interpreted as an index of autonomic arousal rather than emotional valence or memory strength. Thus, results are discussed strictly in terms of arousal responses.

BVP was recorded to allow triangulation with cardiovascular indices and potential future computation of heart-rate variability metrics. However, to maintain focus, only GSR-based variables are analyzed in the present study.”

Comment 4: ”Finally, regarding the paper’s structure and logic, the correspondence between the research questions, objectives, and hypotheses is not explicitly clear. It is recommended that the introduction or methods section present these elements clearly in a table or diagram. The discussion section is somewhat lengthy, with some content repeating the results. It should be streamlined to focus more sharply on theoretical interpretation and practical implications.”

Response 4: We thank the reviewer for this valuable observation. Thank you for pointing out the need to make the correspondence between research questions, objectives, and hypotheses explicit. In the revised manuscript, this alignment is clearly presented in Figure 1 (“Structure of research design: from questions to hypotheses”), which maps RQs (R1–R4) to Objectives (O1–O4) and Hypotheses (H1–H4), together with the associated variables and planned analyses. We also added explicit cross-references in the Introduction and Methods to direct readers to Figure 1

Sincerely,

Authors

Round 2

Reviewer 2 Report

Comments and Suggestions for Authors

Comments to the Author

Dear author(s),

Overall, this study addresses a topic of practical significance, employs a sound research design, and presents robust data analysis, providing valuable physiological evidence and nuanced perspectives for the fields of sustainable communication and consumer behavior. However, there remains room for improvement in several areas, including theoretical depth, sample representativeness, the rigor of conclusions, and practical implications. Below are specific recommendations for revision:

First, in the Theory and Research Hypotheses section, it is recommended to strengthen the theoretical foundation. The current explanation of the “emotion-cognition-behavior” pathway appears somewhat underdeveloped. Introducing a more explanatory theoretical framework would be beneficial.

Second, in the Sample and Methods section, the research limitations should be further clarified. Although the stimuli used in the experiment (AI-generated images) facilitate variable control, their ecological validity may be limited. It is advisable to acknowledge this in the limitations and suggest that future studies incorporate real advertising materials to enhance external validity.

Third, in the Discussion and Managerial Implications section, it is recommended to expand the scope of the theoretical contributions and practical applications. The current discussion focuses primarily on marketing communication. The findings could be further connected to the field of organizational behavior, exploring their potential application in contexts such as internal sustainability communication, employee behavior change, and corporate social responsibility (CSR) dissemination, thereby enhancing the paper’s interdisciplinary impact. Furthermore, the existing managerial recommendations are somewhat general and could be made more concrete.

Finally, regarding the overall structure and language expression, the transitions between some paragraphs could be made smoother and more natural. Adding appropriate transitional sentences would improve logical coherence. Additionally, it is recommended to perform a consistent check of figure/table numbering and citation formatting throughout the manuscript.

Best regards.

Author Response

Response to Reviewer Comments

1. Summary

Thank you very much for your thorough review of this manuscript. Below, we provide detailed responses to each of your comments, with corresponding revisions and corrections highlighted in the re-submitted files using track changes.

2. Questions for General Evaluation

Reviewer’s Evaluation

Response and Revisions

Is the content succinctly described and contextualized with respect to previous and present theoretical background and empirical research (if applicable) on the topic?

Can be improved

Please, see the responses below.

Are the research design, questions, hypotheses and methods clearly stated?

Can be improved

Are the arguments and discussion of findings coherent, balanced and compelling?

Can be improved

For empirical research, are the results clearly presented?

Can be improved

Is the article adequately referenced?

Can be improved

Are the conclusions thoroughly supported by the results presented in the article or referenced in secondary literature?

Can be improved

3. Point-by-point response to Comments and Suggestions for Authors

Comment 1. ”Overall, this study addresses a topic of practical significance, employs a sound research design, and presents robust data analysis, providing valuable physiological evidence and nuanced perspectives for the fields of sustainable communication and consumer behavior. However, there remains room for improvement in several areas, including theoretical depth, sample representativeness, the rigor of conclusions, and practical implications. Below are specific recommendations for revision:

First, in the Theory and Research Hypotheses section, it is recommended to strengthen the theoretical foundation. The current explanation of the “emotion-cognition-behavior” pathway appears somewhat underdeveloped. Introducing a more explanatory theoretical framework would be beneficial.”

Response 1. Thank you very much for this valuable comment. We agree that the theoretical development of the “emotion–cognition–behavior” pathway required deepening. In the revised manuscript, we have strengthened the theoretical grounding by: 1) expanding the justification of the model through recent literature on affective processing and decision-making; and 2) integrating dual-processing theory and appraisal theory, which explain more comprehensively how emotional stimuli shape cognitive evaluations and subsequently behavioral outcomes in sustainable consumption contexts. These additions are now reflected in Section 2 (Theory and Research Hypotheses). We believe the enhanced theoretical structure better supports the hypotheses and improves the explanatory coherence of the study.

Comment 2. ”Second, in the Sample and Methods section, the research limitations should be further clarified. Although the stimuli used in the experiment (AI-generated images) facilitate variable control, their ecological validity may be limited. It is advisable to acknowledge this in the limitations and suggest that future studies incorporate real advertising materials to enhance external validity.”

Response 2: Thank you for highlighting this important methodological point. We agree that although AI-generated visual stimuli allow for tight variable control, they may reduce ecological validity compared with naturally occurring advertising materials. In the revised version, we have acknowledged this limitation explicitly in limitations and we have suggested that future studies could employ real-world advertisements and/or field experiments to enhance external validity and test whether the observed effects persist in more naturalistic settings.

Comment 3. ”Third, in the Discussion and Managerial Implications section, it is recommended to expand the scope of the theoretical contributions and practical applications. The current discussion focuses primarily on marketing communication. The findings could be further connected to the field of organizational behavior, exploring their potential application in contexts such as internal sustainability communication, employee behavior change, and corporate social responsibility (CSR) dissemination, thereby enhancing the paper’s interdisciplinary impact. Furthermore, the existing managerial recommendations are somewhat general and could be made more concrete.”

Response 3. We appreciate this insightful suggestion. In response, we have broadened the Discussion and Managerial Implications section. Specifically:

  • The theoretical implications now link our findings to the organizational behavior literature, particularly internal sustainability communication, employee engagement, and CSR messaging.
  • The managerial implications have been revised to include more concrete, actionable recommendations, such as guidelines for sustainable message framing, internal communication strategies to reinforce pro-social behavior among employees, and visual communication practices for CSR programs.

These updates can be found in Section of Discussion and Section dedicated to managerial  Implications.

Comment 4. ”Finally, regarding the overall structure and language expression, the transitions between some paragraphs could be made smoother and more natural. Adding appropriate transitional sentences would improve logical coherence. Additionally, it is recommended to perform a consistent check of figure/table numbering and citation formatting throughout the manuscript.”

Response 5. Thank you for pointing this out. We have carried out a thorough revision of the manuscript to improve logical flow, and transitional sentences have been added where necessary to ensure smoother connections between paragraphs. In addition, we have conducted a full consistency check of figure and table numbering, in-text citations, and reference formatting in accordance with the journal’s guidelines. We trust these revisions enhance clarity and readability.

Sincerely,

Authors
